# Treatment-Related Adverse Events in Extended Stage Small Cell Lung Cancer Patients Receiving First-Line Chemoimmunotherapy Versus Chemotherapy Alone: A Systematic Review and Meta-Analysis

**DOI:** 10.3390/cancers17091571

**Published:** 2025-05-05

**Authors:** Elsa Vitale, Alessandro Rizzo, Lorenza Maistrello, Deniz Can Guven, Omar Cauli, Domenico Galetta, Vito Longo

**Affiliations:** 1Medical Thoracic Oncology Unit, IRCCS Istituto Tumori “Giovanni Paolo II”, 70124 Bari, Italy; e.vitale@oncologico.bari.it (E.V.); galetta@oncologico.bari.it (D.G.); 2Struttura S.S.D.C.O.r.O., Bed Management Presa in Carico, TDM, IRCCS Istituto Tumori “Giovanni Paolo II”, 70124 Bari, Italy; a.rizzo@oncologico.bari.it; 3IRCCS San Camillo Hospital, 30121 Venice, Italy; lorenza.maistrello@hsancamillo.it; 4Medical Oncology Clinic, Health Sciences University, Elazig City Hospital, Elazig 23280, Turkey; denizcguven@hotmail.com; 5Nursing Department, Faculty of Nursing and Podiatrics, Universitat de València, 46010 Valencia, Spain; omar.cauli@uv.es

**Keywords:** adverse events, chemotherapy, chemoimmunotherapy, patient, small cell lung cancer, safety

## Abstract

Nowadays the prognosis of Extended Stage (ES)–Small cell lung cancer (SCLC) patients is poor. However, a high response rate to first-line chemotherapy (CT) and the addition of immune checkpoint inhibitors (ICIs) have notably ameliorated the outcome of these patients. The aim of our study is to compare treatment-related adverse events (TRAEs) between ES-SCLC patients receiving first-line ICIs added to CT and those receiving CT alone. Our analysis confirms the manageable safety profile of CT, this remains an issue to be further investigated.

## 1. Introduction

Small cell lung cancer (SCLC) encounters nearly 14–17% of all lung cancer cases [1]. It is a very aggressive cancer type with neuroendocrine malignancy and a high growth fraction, genomic instability, and a prompt time to doubling, evolving in advanced growth of metastases and a consequent bleak prognosis. Evidence highlights that nearly 70% of diagnosed cases already present advanced disease. Thus, the remarkable advances in treatment with first-line chemotherapy (CT) did not ameliorate the prognosis of SCLC patients, recording a 5-year survival rate of about 5% [2]. Immune checkpoint inhibitors (ICIs) have reported excellent results against several cancer typologies and thus, they have been approved by worldwide regulatory agencies. The improvement of these combined therapies includes ICIs and other drugs that actively act in solid tumors such as small cell lung cancer, often diagnosed at the latest phases. The use of ICI in monotherapy has been associated with a low incidence of organ-specific severe adverse events, unlike traditional chemotherapies and multikinase inhibitors, but the adverse effects of chemotherapy-associated immunotherapy in this tumor type are still unknown. The advent of combined therapies also including ICIs and other anticancer drugs with different approaches of action updated knowledge about security risks, and opportune handling approaches referred to the combination of these drugs; however, it requires attention to centralized treatment and thus, closely monitoring the occurrence of potentially lethal side effects. However, a comprehensive differentiation of the prevalence of clinically significant adverse effects between chemoimmunotherapy compared to only chemotherapy in extended-stage small cell lung cancer is necessary. Recently, adding immune checkpoint inhibitors (ICIs) to first-line CT has notably ameliorated the results of Extended Stage (ES)–SCLC patients, since the usual first-line treatment for extensive-stage SCLC is platinum CT [3]. SCLC has a great mutation rate, which could respond to ICIs and ameliorate outcomes in these patients [4,5]. In particular, the phase III IMpower133 trial first reported overall survival (OS) improvement thanks to the addition of atezolizumab, an anti-PD-L1, to platinum-based CT [6,7]. Later, the phase III Caspian trial, using first-line anti-PD-L1 durvalumab mixed with CT, confirmed these results [8]. More recently, three other phase III trials, like CAPSTONE-1, ASTRUM005, and RATIONALE-312 supported the advantage of ICIs adding with CT in the first-line setting, specifically adding the anti-PD-L1 adebrelimab [9], the anti-PD1 serplulimab [10], and the anti-PD1 tislelizumab to CT as first-line [11]. However, in NSCLC, biomarkers predictive of response to immune-based treatments are lacking in SCLC [12]. On the other hand, the administration of ICIs is linked to a broad variety of toxicities, that are distinct from the toxicities of CT and targeted therapies. The unique toxicity profile of ICIs depends on their mechanisms of action, requiring specific management [13]. In some cases, immunotherapy toxicity can be severe and even life-threatening [13]. Furthermore, considering the absence of biomarkers predictive of immunotherapy response for SCLC, some patients do not benefit from ICIs but at the same time experience toxicity.

Based on these premises, we performed a systematic review and meta-analysis to compare TRAEs between ES-SCLC patients receiving first-line ICIs adding CT and those receiving only CT among all phase III clinical trials available in the current literature.

## 2. Materials and Methods

### 2.1. Search Strategy

The present systematic review and meta-analysis were conducted thanks to the Equator Network checklists, particularly thanks to the Preferred Reporting Items for Systematic Reviews and Meta-analysis (PRISMA) reporting system [14]. The protocol was registered with PROSPERO no. CRD42024563686.

All phase III clinical trials published between 15 June 2008, and 30 June 2024, comparing ICIs plus systemic CT versus CT alone in treatment-naïve ES-SCLC patients were retrieved by three different authors. The search keywords used in PubMed/Medline, Cochrane Library, and EMBASE were the following: “double-blind” AND “phase III” AND “immunotherapy” AND “SCLC” OR “ES-SCLC”. Only articles published in peer-reviewed journals and written in the English language were included.

### 2.2. Selection Criteria

Inclusion criteria covered: (1) prospective phase III randomized controlled trials (RCTs) in ES-SCLC; (2) patients treated with ICIs and systemic CT comparing to only CT; (3) studies contained data referring to any grade TRAEs, and grade 3 or more TRAEs. In clinical trials including two arms, safety data were combined, and the results were matched with those of the CT alone group.

### 2.3. Data Extraction

During the first phase of the present systematic review and meta-analysis, a total of 17 studies were found. Eleven studies were excluded, as duplicates and 6 met the inclusion criteria [7,9,10,11,15,16]. The article selection procedure was displayed in the Prisma flow diagram (Figure 1). Data were obtained from each publication: (1) general information (author, year, phase, country of origin); (2) interventions and dosage; (3) number of participants; (4) primary outcomes; (5) numbers of any grade and grade 3 or more TRAEs.

### 2.4. Data Analysis

The primary outcomes of this meta-analysis were incidence (defined as the number of patients with at least one TRAEs) divided by the total number of patients analyzed for the specific event). For each type of TRAE, effect size was calculated by estimating Odds Ratios (OR) using the Mantel-Haenszel method (without continuity correction) [17] OR were calculated for any grade of TRAE and for grade ≥ 3 in both the Immunotherapy and Placebo groups of patients. Subsequently, the TRAEs were subdivided according to their type, namely: blood abnormalities not yet specified, blood pressure alterations, gastrointestinal events, general disorders, and administration site conditions, haematological events, hepatic function, metabolism, and nutrition disorders, respiratory thoracic and mediastinal disorders, skin and subcutaneous disorders and thyroid disorders.

Two separate subgroup meta-analyses were performed, one for any grade and one for grade > 3, considering the stratification of adverse events by type.

In every meta-analysis, heterogeneity between TRAEs was assessed by calculating the total Cochrane Q test (Q). A significant Q value (P_Q_ < 0.05) suggests the presence of significant heterogeneity between TRAEs. Additionally, the Tau-squared coefficient (tau^2^) and the I-squared inconsistency index (I^2^) were used to estimate the percentage of true variability in the observed effects [18,19]. Degrees of heterogeneity were considered as follows: I^2^ values of 25%, 50%, and 75% suggest low, moderate, and high degrees of heterogeneity, respectively.

Because we expected there to be a high degree of heterogeneity among TRAEs, a random-effects pooling model with the Knapp-Hartung adjustment method (HKSJ) was assessed to group the effect sizes [20]. Then, the influence analysis based on the Baujat Plot and Leave-One-Out method using Viechtbauer and Cheung’s cut-off approach was performed [21] to identify the presence of influential cases and outliers. When outliers were identified, they were removed from the analyses.

A forest plot was obtained for each meta-analysis [22]. Publication bias was assessed using a contour funnel plot. Statistical analysis was performed using R software (version 4.3.2) and the significance level of the analysis was established at *p* < 0.05.

## 3. Results

All data were processed according to different severity grades, such as “any grade” and grade 3 or more [20].

### 3.1. The Main Features of the Included Any Grade TRAEs

Twenty-six types of adverse events were included, grouped into ten categories, for a total of 43,391 observations (observations in immune group n = 22,643 and in placebo group n = 20,748) and 9831 events (Table 1).

### 3.2. Meta-Analysis

Since the Cochrane Q test displays great heterogeneity (P_Q_ < 0.0001; tau^2^ = 0.045 and I^2^ = 73.80% with 95% CI = [61.4%; 82.2%]), we assessed an influence analysis that highlighted 3 cases to be persuasive (Rash, Hyponatremia, and Hypothyroidism). Thanks to the Gosh analysis, Rash, Hyponatremia, and Hypothyroidism were excluded as outliers. The removal of the outliers leads to a high decrease in heterogeneity indices (P_Q_ = 0.392; tau^2^ = 0.002; I^2^ = 4.9% with 95%-CI = [0.0%; 36.1%]) (Figure 2). From the results of the subgroup meta-analysis, we can see that there is a difference in the effect of therapy between the groups with regard to adverse events related to blood pressure alterations and hematological events.

The Funnel Plot (Figure 3) identified the absence of publication bias.

The trim and fill approach was performed to recognize asymmetry and identify potential bias. The grey dots represent the studies included in the meta-analysis. The colors of the funnels suggest the significance levels. On the other hand, the white area, suggests the no significant area (i.e., where *p* > 0.1). When there is no publication bias, all studies would lie symmetrically around our pooled effect size (the striped line) within the form of the funnel.

### 3.3. The Main Features of the Included Grade 3 or More TRAEs

Of the 26 types of adverse events included, only 25 types were taken into account, as ORs could not be calculated for the adverse event “Hypoalbuminemia”. The adverse events included were grouped into 10 categories, for a total of 108,867 observations (observations in immune group n = 55,528 and in placebo group n = 53,339) and 3388 events (Table 2).

### 3.4. Meta-Analysis

By performing Gosh analysis, three influential cases (i.e., Hyponatremia, Hypothyroidism, and Rash) were identified and then removed as outliers.

Cochrane’s Q test showed that there was no heterogeneity (P_Q_ = 0.549; tau^2^ = 0; I^2^ = 0% with 95%-CI = [0.0%; 46.2%]). A statistically significant difference in terms of grade 3 or more hypertension was observed in ES-SCLC patients receiving CT alone compared with those treated with ICIs plus CT was observed (Figure 4).

The Funnel Plot (Figure 5) indicates that no publication bias is present.

The trim and fill approach was performed to recognize asymmetry and identify potential bias. The grey dots represent the studies included in the meta-analysis. The colors of the funnels suggest the significance levels. On the other hand, the white area, suggests the no significant area (i.e., where *p* > 0.1). When there is no publication bias, all studies would lie symmetrically around our pooled effect size (the striped line) within the form of the funnel.

## 4. Discussion

The present systematic review and meta-analysis aimed to compare TRAEs between ES-SCLC patients receiving first-line ICIs plus CT and those receiving CT alone in phase III clinical trials. As regards all grade TRAEs [23], a total of 26 types of adverse events were reported, which were grouped into ten categories; our analysis suggested a statistically significant difference in terms of hematological events, which were more commonly observed in ES-SCLC patients receiving ICIs plus CT compared with CT alone. However, no difference was found regarding grade 3 or higher hematological side effects, in particular no difference regarding febrile neutropenia. Conversely, blood pressure alterations such as hypertension were more frequent in patients treated with CT alone. As regards grade 3 or more TRAEs [23], a total of 26 types of adverse events were encountered and only 25 types were taken into account, as ORs could not be calculated for the adverse event Hypoalbuminemia. All the 25 types were sub-grouped into 10 categories; as in the case of all grade TRAEs, a statistically significant difference in terms of grade 3 or more hypertension was observed in ES–SCLC patients receiving CT alone compared with those treated with ICIs plus CT. The higher rate of hypertension in patients treated with CT alone could be partly due to lower corticosteroid exposure during ICIs plus CT. The advent of ICIs and immune-based combinations has improved the prognosis of patients with ES–SCLC [24]. Recently, chemoimmunotherapy has emerged as a first-line treatment in this setting, demonstrating efficacy in multiple phase III trials and leading to practice-changing clinical benefits [25]. In addition, several studies are ongoing to determine the role of other, emerging combinations and sequences of immunotherapy with other anticancer treatments. However, the inhibition of immune checkpoints can increase the incidence of side effects, that may affect several organs or tissues, including—among the others—skin, colon, endocrine organs, liver, and lungs [26]. Despite previous studies having suggested a possible correlation between TRAEs and improved survival outcomes in SCLC patients, this association has not been validated yet [27]. Suitable management of TRAEs appears crucial to maximize treatment goals and to simultaneously identify patients’ vulnerability to TRAEs. Previous studies have explored the role of other factors that could play a role in TRAEs, such as low neutrophil-to-lymphocyte ratio (NLR), high body mass index (BMI), high serum albumin (Alb) level, and low Eastern Cooperative Oncology Group (ECOG) performance status (PS) [28,29,30]. However, very few evidence dealt with chemoimmunotherapy and its related TRAEs [31,32]. All these clinical symptoms can negatively affect the quality of life of cancer patients [33], and thus, the study of their incidences and related prompt management is fundamental. Close monitoring and a multidisciplinary approach are essential to managing these TRAEs effectively, allowing patients to continue benefiting from immunotherapy. In this meta-analysis, we have identified 3 cases as outliers identifying patients with Rash, Hyponatremia, and Hypothyroidism to be particularly sensitive to Immunotherapy-adverse effects. It is well established that non-specific immune activation can lead to immune-related adverse effects, with the skin and its appendages being the most common goals [34]. PD-1/PD-L1 inhibitors are often well tolerated, though some of the most frequent immune-related adverse events are cutaneous rash or pruritus [35]. In the latest clinical trials, more than 40% of patients experienced some type of cutaneous signs [35], including several inflammatory reactions, like maculopapular rash, pruritus, and psoriasiform and lichenoid eruptions. Immune-related cutaneous adverse effects occur early, with maculopapular rash occurring within the first 6 weeks after the initial dose of immune checkpoint inhibitor [36]. The severity of the rash could be amplified by concomitant therapies such as kinase inhibitors. A recent meta-analysis aiming to analyze any adverse events induced by immunotherapy, including dermatological events, reported a significantly higher RR for the incidence of rash when PD-1/PD-L1 inhibitors were associated with kinase inhibitors compared with PD-1/PD-L1 inhibitor monotherapy [37]. In extended stage small cell lung cancer patients, adding PD-1/PD-L1 inhibitors to chemotherapy has already been reported to lead to a higher incidence of hyponatremia, and hypothyroidism [38]. The accurate processes inducing hyponatremia in NSCLC patients are still unknown. Syndrome of inappropriate antidiuresis represents around 2–4% of the cases of hyponatremia in NSCLC [39]. The presence of brain metastases and concomitant treatment with mannitol as well as the use of anti-epileptic drugs, characterized by impaired (i.e., reduced) water loss, could be the cause of SIAD in NSCLC patients. Moreover, the presence of comorbidities, including heart or kidney failure, gastrointestinal leakage, and cancer treatment, or concomitant medications as per opioid drugs, may contribute to sodium loss [40]. In addition, immune checkpoint inhibitors can rarely induce severe hyponatriemia by inappropriate antidiuretic hormone ADH release in predisposed patients with Mulvihill-Smith Syndrome, a rare genetic syndrome that has been linked to metabolic dysfunctions and early-onset tumors and malignancies [41]. Jaafar et al. 2018 [32] reviewed the incidence of hypothyroidism induced by PD-1/PD-L1 inhibitors. The mechanism of PD-1/PD-L1-induced dysthyroidism (both hypo- and hyperthyroidism) is still unknown, also due to the lack of studies in this regard. It has been hypothesized that the presence of antithyroid antibodies such as TPO and thyroglobulin antibodies could be a favorable factor leading to hypothyroidism. A prospective study of 51 patients with NSCLC treated with pembrolizumab [42], showed that thyroid dysfunction was closely associated with thyroid antibodies and that 80% of patients with thyroiditis had thyroid antibodies, suggesting a modulatory effect of PD-1 antibodies on humoral immunity.

It should also keep in mind that hypothyroidism and hyponatremia may have a causal relationship. Although is a rare comorbid condition, hypothyroidism can be one of the causes of hyponatremia [43]. The main mechanism for the development of hyponatremia in patients with chronic hypothyroidism is the decreased capacity of free water excretion due to elevated antidiuretic hormone levels, which are mainly attributed to the hypothyroidism-induced decrease in cardiac output [43,44,45].

Our meta-analysis has some strengths and limitations to be considered. Surely, our study embraced all the most updated RCTs and their related data in TRAEs incidence. However, some limitations should be underlined. First, the current meta-analysis was based on pooled data, and thus, confounding factors and single-patient variables (e.g., patient age, comorbidities, concomitant medications, etc.) were not assessed. Second, in all the screened RCTs patients were treated with heterogeneous treatments. Thus, this element could have produced some bias affecting our results. The presence of selection bias cannot be excluded. Moreover, our study did not reveal any differences regarding endocrine TRAEs, but immunotherapy is strongly associated with hypothyroidism in several cancer types [46]. Although this meta-analysis is unable to show a difference in endocrine events, several real-world studies have reported increased hypothyroidism even in SCLC patients treated with ICI plus CT [47,48]. On the other hand, this was the first literature review that deeply analyzed each TRAE encountering frequency for each one.

## 5. Conclusions

Surely, further studies are needed to better highlight in which direction healthcare professionals should address the management of TRAEs in this setting. The safety of combinatorial treatments with CT plus ICIs remains an issue in medicine, and despite our analysis further confirms the manageable safety profile of chemoimmunotherapy, it is important to remain vigilant for TRAEs in ES-SCLC patients receiving these anticancer treatments [49,50]. Finally, we did not obtain statistically significant results either by considering the overall analyses or in the analyses stratified by specific irAE, either Any Grade or Grade >3, however, since some recent studies suggest that the minimal clinically important difference (MCID) should also be considered [51], this will be explored in the future.

## Figures and Tables

**Figure 1 cancers-17-01571-f001:**
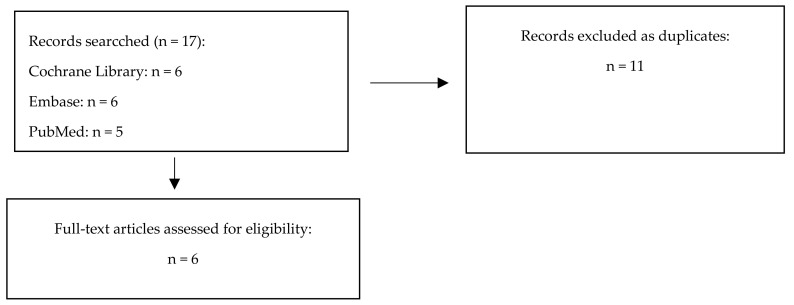
The PRISMA flow chart.

**Figure 2 cancers-17-01571-f002:**
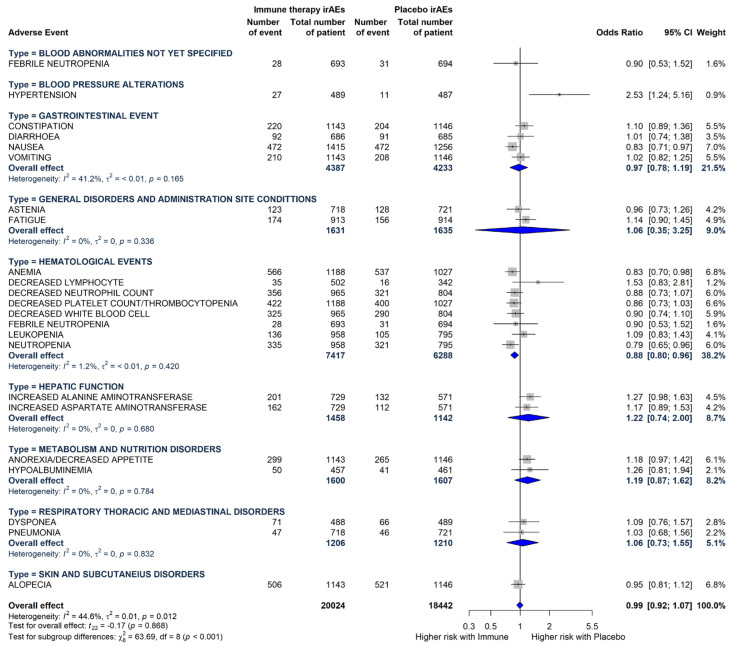
Odds ratios and 95% CIs for each type of TRAE derived from pooled estimates in the comparison between the immunotherapy plus chemotherapy arm and the chemotherapy alone arm.

**Figure 3 cancers-17-01571-f003:**
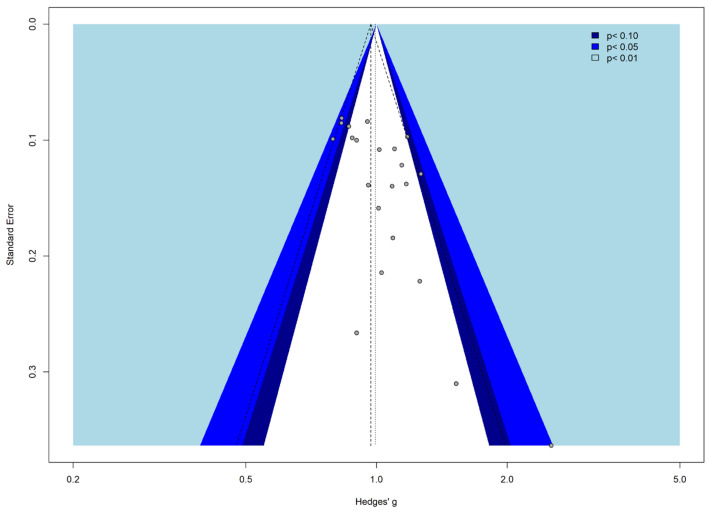
Contour-enhanced Funnel Plot for publication bias.

**Figure 4 cancers-17-01571-f004:**
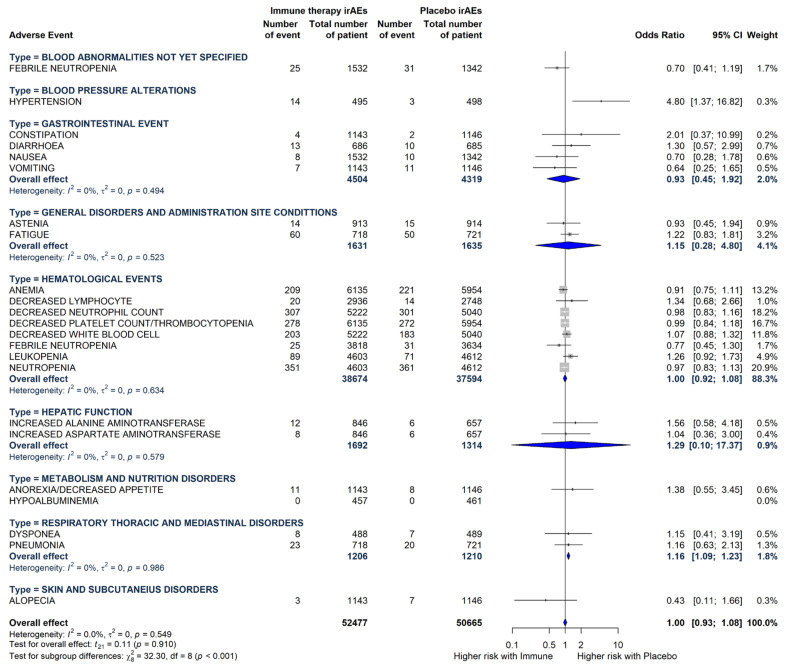
Odds ratios and 95% CIs for each grade 3 or more TRAEs derived from pooled estimates in the comparison between the immunotherapy plus chemotherapy arm and the chemotherapy alone arm.

**Figure 5 cancers-17-01571-f005:**
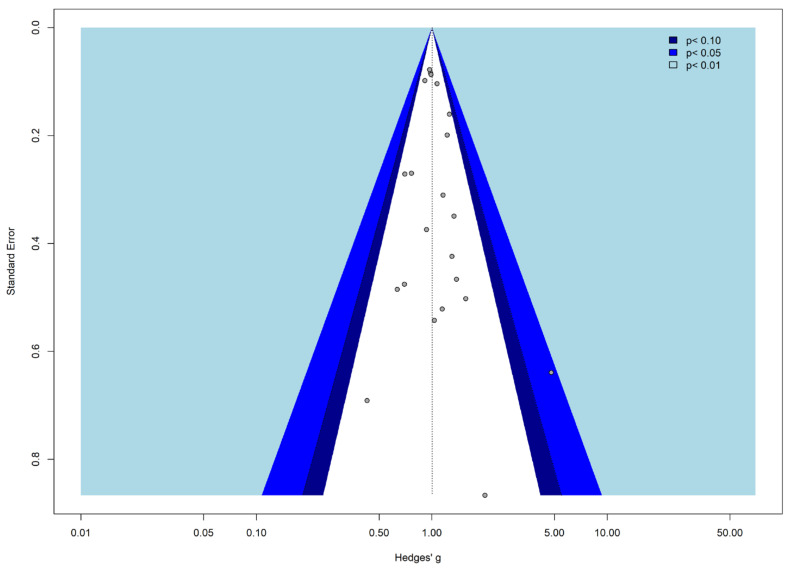
Contour-enhanced Funnel Plot for publication bias.

**Table 1 cancers-17-01571-t001:** Incidence of individual Any Grade TRAEs between the immunotherapy plus chemotherapy arm and the chemotherapy alone arm.

Adverse Events	Immunotherapy			Chemotherapy Alone		
	Events (n)	Patients (n)	Incidence: Events (n)/Patients (n) (%)	Events (n)	Patients (n)	Incidence: Events (n)/Patients (n) (%)
**BLOOD ABNORMALITIES NOT YET SPECIFIED**						
FEBRILE NEUTROPENIA	28	693	0.04	31	694	0.04
HYPONATREMIA	109	1217	0.09	65	1060	0.06
**BLOOD PRESSURE ALTERATIONS**						
HYPERTENSION	27	489	0.06	11	487	0.02
**GASTROINTESTINAL EVENT**						
DIARRHOEA	92	686	0.13	91	685	0.13
NAUSEA	472	1415	0.33	472	1256	0.38
VOMITING	210	1143	0.18	208	1146	0.18
CONSTIPATION	220	1143	0.19	204	1146	0.18
**GENERAL DISORDERS AND ADMINISTRATION SITE CONDITTIONS**						
ASTENIA	123	718	0.17	128	721	0.18
FATIGUE	174	913	0.19	156	914	0.17
**HEMATOLOGICAL EVENTS**						
DECREASED WHITE BLOOD CELL	325	965	0.34	290	804	0.36
DECREASED NEUTROPHIL COUNT	356	965	0.37	321	804	0.40
NEUTROPENIA	335	958	0.35	321	795	0.40
ANEMIA	566	1188	0.48	537	1027	0.52
LEUKOPENIA	136	958	0.14	105	795	0.13
FEBRILE NEUTROPENIA	28	693	0.04	31	694	0.04
DECREASED PLATELET COUNT/THROMBOCYTOPENIA	422	1188	0.36	400	1027	0.39
DECREASED LYMPHOCYTE	35	502	0.07	16	342	0.05
**HEPATIC FUNCTION**						
INCREASED ALANINE AMINOTRANSFERASE	201	729	0.28	132	571	0.23
INCREASED ASPARTATE AMINOTRANSFERASE	162	729	0.22	112	571	0.20
**METABOLISM AND NUTRITION DISORDERS**						
ANOREXIA/DECREASED APPETITE	299	1143	0.26	265	1146	0.23
HYPOALBUMINEMIA	50	457	0.11	41	461	0.09
**RESPIRATORY THORACIC AND MEDIASTINAL DISORDERS**						
PNEUMONIA	47	718	0.07	46	721	0.06
DYSPONEA	71	488	0.15	66	489	0.13
**SKIN AND SUBCUTANEIUS DISORDERS**						
ALOPECIA	506	1143	0.44	521	1146	0.45
RASH	58	450	0.13	31	452	0.07
**THYROID DISORDERS**						
HYPOTHYROIDISM	138	952	0.14	40	794	0.05

**Table 2 cancers-17-01571-t002:** Incidence of individual grade 3 or more TRAEs between the immunotherapy plus chemotherapy arm and the chemotherapy alone arm.

Adverse Events	Immunotherapy			Chemotherapy Alone		
	Events (n)	Patients (n)	Incidence: Events (n)/Patients (n) (%)	Events (n)	Patients (n)	Incidence: Events (n)/Patients (n) (%)
**BLOOD ABNORMALITIES NOT YET SPECIFIED**						
FEBRILE NEUTROPENIA	25	1532	0.02	31	1342	0.02
HYPONATREMIA	28	1532	0.02	18	1342	0.01
**BLOOD PRESSURE ALTERATIONS**						
HYPERTENSION	14	495	0.03	3	498	0.01
**GASTROINTESTINAL EVENT**						
DIARRHOEA	13	686	0.02	10	685	0.01
NAUSEA	8	1532	0.01	10	1342	0.01
VOMITING	7	1143	0.01	11	1146	0.01
CONSTIPATION	4	1143	0	2	1146	0
**GENERAL DISORDERS AND ADMINISTRATION SITE CONDITTIONS**						
ASTENIA	14	913	0.02	15	914	0.02
FATIGUE	60	718	0.08	50	721	0.07
**HEMATOLOGICAL EVENTS**						
DECREASED WHITE BLOOD CELL	203	5222	0.04	183	5040	0.04
DECREASED NEUTROPHIL COUNT	307	5222	0.06	301	5040	0.06
NEUTROPENIA	351	4603	0.08	361	4612	0.08
ANEMIA	209	6135	0.03	221	5954	0.04
LEUKOPENIA	89	4603	0.02	71	4612	0.02
FEBRILE NEUTROPENIA	25	3818	0.01	31	3634	0.01
DECREASED PLATELET COUNT/THROMBOCYTOPENIA	278	6135	0.05	272	5954	0.05
DECREASED LYMPHOCYTE	20	2936	0.01	14	2748	0.01
**HEPATIC FUNCTION**						
INCREASED ALANINE AMINOTRANSFERASE	12	846	0.01	6	657	0.01
INCREASED ASPARTATE AMINOTRANSFERASE	8	846	0.01	6	657	0.01
**METABOLISM AND NUTRITION DISORDERS**						
ANOREXIA/DECREASED APPETITE	11	1143	0.01	8	1146	0.01
HYPOALBUMINEMIA	0	457	0	0	461	0
**RESPIRATORY THORACIC AND MEDIASTINAL DISORDERS**						
PNEUMONIA	23	718	0.03	20	721	0.03
DYSPONEA	8	488	0.02	7	489	0.01
**SKIN AND SUBCUTANEIUS DISORDERS**						
ALOPECIA	3	1143	0	7	1146	0.01
RASH	8	450	0.02	0	452	0
**THYROID DISORDERS**						
HYPOTHYROIDISM	2	1069	0	0	880	0

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
