# Peer review of "Treatment-Related Adverse Events in Extended Stage Small Cell Lung Cancer Patients Receiving First-Line Chemoimmunotherapy Versus Chemotherapy Alone: A Systematic Review and Meta-Analysis"

_cancers, 2025, doi:10.3390/cancers17091571_

Round 1
Reviewer 1 Report
Comments and Suggestions for Authors
The manuscript titled "Treatment-related adverse events in extended stage small cell lung cancer patients receiving first-line chemoimmunotherapy versus chemotherapy alone: A systematic review and meta-analysis" presents a well-structured analysis of the safety profile of chemoimmunotherapy in extended-stage small cell lung cancer (ES-SCLC) patients. Given the increasing clinical relevance of immune checkpoint inhibitors (ICIs) in lung cancer treatment, this work contributes to the comparative safety of chemoimmunotherapy versus chemotherapy alone.
However, some minor revisions are necessary before considering the manuscript for publication.
1) The authors excluded rash, hyponatremia, and hypothyroidism as outliers. However, their exclusion suggests that these adverse events show the most significant differences between the treatment groups. In addition to just removing them to improve statistical consistency, the authors should discuss these specific adverse events in more detail. Their distinct behavior might reveal missed insights into the differential toxicity of chemoimmunotherapy.
2) To strengthen the discussion, the manuscript should incorporate possible physiological and molecular mechanisms underlying the adverse events identified as outliers. Providing mechanistic explanations will enhance the biological relevance of the findings and offer a clearer interpretation of the manuscript's findings.
3) The resolution and clarity of the forest plots need to be improved. Higher-resolution images should be used to improve readability.
4) The tables are difficult to navigate. Consider restructuring them into a horizontal format to facilitate comparison across different treatment arms. Adjusting column headers for better readability would enhance the accessibility of the data.
5) The funnel plots are appropriately used to assess publication bias; however, their interpretation is not immediately intuitive. Adding explanatory notes to the figure legends would be beneficial. For example, indicating what each point represents and explaining the significance of the blue areas.
Author Response
Rebuttal letter
Treatment-related adverse events in extended stage small cell lung cancer patients receiving first-line chemoimmunotherapy versus chemotherapy alone: A systematic review and meta-analysis
We thank both Editor and reviewers for their valuable comments.
Changes and modifications have been marked in red in the text.
Reviewer Comments:
Reviewer 1
The manuscript titled "Treatment-related adverse events in extended stage small cell lung cancer patients receiving first-line chemoimmunotherapy versus chemotherapy alone: A systematic review and meta-analysis" presents a well-structured analysis of the safety profile of chemoimmunotherapy in extended-stage small cell lung cancer (ES-SCLC) patients. Given the increasing clinical relevance of immune checkpoint inhibitors (ICIs) in lung cancer treatment, this work contributes to the comparative safety of chemoimmunotherapy versus chemotherapy alone.
However, some minor revisions are necessary before considering the manuscript for publication.
R1. The authors excluded rash, hyponatremia, and hypothyroidism as outliers. However, their exclusion suggests that these adverse events show the most significant differences between the treatment groups. In addition to just removing them to improve statistical consistency, the authors should discuss these specific adverse events in more detail. Their distinct behavior might reveal missed insights into the differential toxicity of chemoimmunotherapy.
A1. No, the exclusion criteria is not related to the number of cases. Initially, these studies were considered for the calculation of heterogeneity, but subsequent analyses regarding the presence of influential cases (and their nature), showed that these were outliers and, consequently, were excluded from the final meta-analysis because they would otherwise invalidate the results of the entire meta-analysis, which would be found to be unreliable.
R2. To strengthen the discussion, the manuscript should incorporate possible physiological and molecular mechanisms underlying the adverse events identified as outliers. Providing mechanistic explanations will enhance the biological relevance of the findings and offer a clearer interpretation of the manuscript's findings.
A2. According to Reviewer’ suggestion we have added in the Discussion section the following possible physiological and molecular mechanisms underlying the adverse events identified as outliers:
“In this meta-analysis, we have identified 3 cases as outliers identifying patients with Rash, Hyponatremia and Hypothyroidism to be particular sensitive to Immunotherapy-adverse effects. It is well established that non-specific immune activation can lead to immune-related adverse effects, with the skin and its appendages being the most frequent targets (Geisler et al., 2020). PD-1/PD-L1 inhibitors are generally well tolerated, though some of the most common immune-related adverse events are cutaneous rash or pruritus (Shen et al. 2018). Up to 40% of patients in recent clinical trials experienced some type of cutaneous manifestation (Shen et al. 2018) include a diverse group of inflammatory reactions, the most frequent subtypes being maculopapular rash, pruritus, and psoriasiform and lichenoid eruptions. Immune-related cutaneous adverse effects occur early, with maculopapular rash occurring within the first 6 weeks after the initial dose of immune checkpoint inhibitor (Geisler et al., 2020). The severity of rash could be amplified by concomitant therapies such as kinase inhibitors. A recent meta-analysis aiming to analyse any adverse events induce by immunotherapy, including dermatological events, reported a significant higher RR for the incidence of rash when PD-1/PD-L1 inhibitors were combined with kinase inhibitors compared with PD-1/PD-L1 inhibitor monotherapy (Inoue et al., 2025). In extended stage small cell lung cancer patients, adding PD-1/PD-L1 inhibitors to chemotherapy is already reported that leads to a higher incidence of hyponatremia, and hypothyroidism (Moraes et al., 2024). The exact mechanisms leading to hyponatremia in NSCLC patients remain unclear. Syndrome of Inappropriate
Syndrome of inappropriate antidiuresis represents around 2–4% of the cases of hyponatremia in NSCLC (Kanaji et al., 2014). The presence of brain metastases and concomitant treatment with mannitol as well as the use of anti-epileptic drugs, characterized by impaired (i.e., reduced) water loss, could be cause of SIAD in NSCLC patients. Moreover, the presence of comorbidities, including heart or kidney failure, gastrointestinal leakage and cancer treatment, or concomitant medications as per opioid drugs, may contribute to the sodium loss (Catalano et a., 2024). In addition, immune checkpoint inhibitors can rarely induce severe hyponatriemia by inappropriate antidiuretic hormone ADH release in predisposed patients with Mulvihill-Smith Syndrome, a rare genetic syndrome that has been associated with metabolic abnormalities and early-onset tumors, including malignancies (Tavdy et al., 2024). Jaafar et al. 2018 reviewed the incidence of hypothyroidism induced by PD-1/PD-L1 inhibitors. The mechanism of PD-1/PD-L1-induced dysthyroidism (both hypo- and hyperthyroidism) is not fully understood, and related studies are scarce. It has been hypothesized that the presence of antithyroid antibodies such as TPO and thyroglobulin antibodies could be a positive predictive factor for developing hypothyroidism, In a prospective study of 51 patients with NSCLC treated with pembrolizumab (Osorio et al., 2017), showed that thyroid dysfunction was closely associated with thyroid antibodies, and that 80% of patients with thyroiditis had thyroid antibodies, suggesting a modulatory effect of PD-1 antibodies on humoral immunity.
It should also keep in mind that hypothyroidism and hyponatremia may have a causal relationship. Although is a rare comorbid condition, hypothyroidism can be one of the causes of hyponatremia (Chu et al. 2023). The main mechanism for the development of hyponatremia in patients with chronic hypothyroidism is the decreased capacity of free water excretion due to elevated antidiuretic hormone levels, which are mainly attributed to the hypothyroidism-induced decrease in cardiac output (Chu et al., 2023; Liamis et al., 2017; Wolf et al., 2017).
R3. The resolution and clarity of the forest plots need to be improved. Higher-resolution images should be used to improve readability.
A3. Figures have been uploaded as separate files to improve their resolutions.
R4. The tables are difficult to navigate. Consider restructuring them into a horizontal format to facilitate comparison across different treatment arms. Adjusting column headers for better readability would enhance the accessibility of the data.
A4. Tables are better adjusted for their readability and accessibility of the data.
R5. The funnel plots are appropriately used to assess publication bias; however, their interpretation is not immediately intuitive. Adding explanatory notes to the figure legends would be beneficial. For example, indicating what each point represents and explaining the significance of the blue areas.
A5. A caption for each funnel plot has been added.
We hope the revised paper will better suit the journal.
The Authors

Reviewer 2 Report
Comments and Suggestions for Authors
This meta-analysis focuses on treatment-related adverse events in extended stage small cell lung cancer patients receiving the first CI and addresses an important issue in clinical medicine. The topic is clinically significant, the number of included studies is appropriate, and the analytical methods are well-suited. The data provided by this manuscript offer valuable insights for clinicians.
1)Please explain in a bit more detail how this topic is clinically significant.
2)Please focus on the results of the main analysis rather than those of the subgroup analysis.
3)Based on Figure4, the text says “a statistically significant difference in terms of grade 3 or more hypertension was observed.” In recent years, data from meta-analyses have been increasingly emphasized. However, reliance solely on the statistical significance of p-values in interpreting such data has been a growing concern. Ideally, both p-values and minimal clinically important difference (MCID) should be considered. Unfortunately, MCID for effect sizes other than mean differences has not been well-established.
Recently, MCID applicable to effect sizes other than mean differences has been proposed as "Horita's MCID." Citing the following reference and incorporating a discussion on this topic, i.e. how you interpret OR=4.80, could further enhance the quality of this manuscript:
Horita N et al. Minimal clinically important difference (MCID) of effect sizes other than mean difference. J Clin Med 2024. DOI 10.69854/jcq.2024.0016.
https://jclinque.com/article/minimal-clinically-important-difference-mcid-of-effect-sizes-other-than-mean-difference
Thank you for the opportunity to review this excellent manuscript.
Author Response
Rebuttal letter
Treatment-related adverse events in extended stage small cell lung cancer patients receiving first-line chemoimmunotherapy versus chemotherapy alone: A systematic review and meta-analysis
We thank both Editor and reviewers for their valuable comments.
Changes and modifications have been marked in red in the text.
Reviewer Comments:
Reviewer 2
This meta-analysis focuses on treatment-related adverse events in extended stage small cell lung cancer patients receiving the first CI and addresses an important issue in clinical medicine. The topic is clinically significant, the number of studies included is appropriate, and the analytical methods are well-suited. The data provided by this manuscript offer valuable insights for clinicians.
R1. Please explain in a bit more detail how this topic is clinically significant.
A1. According to Reviewer’ suggestion we have added on this issue as follows:
“Immune checkpoint inhibitors have demonstrated efficacy against various tumor types and have been approved by several regulatory agencies worldwide. The development of combination therapies that include immunotherapy and other anticancer drugs is actively progressing in poor prognosis solid tumors such as small cell lung cancer, which is often diagnosed at advanced stages. The use of immunotherapy in monotherapy is associated with a relatively low incidence of organ-specific serious adverse events, unlike conventional chemotherapies and multikinase inhibitors, but the adverse effects of chemotherapy-associated immunotherapy in this tumor type are not known. When developing or introducing combination therapies that include immunotherapy and other anticancer drugs with different modes of action, up-to-date knowledge about safety risks and appropriate management strategies based on the profile of the combination drugs are required to be able to personalize oncological treatments and closely monitor the occurrence of potentially lethal side effects. However, a comprehensive comparison of the incidence of clinically significant adverse effects between chemoimmunotherapy versus chemotherapy alone in in extended stage small cell lung cancer.”
R2. Please focus on the results of the main analysis rather than those of the subgroup analysis.
A2. Tables 1 and 2 detail incidence of individual for both “Any Grade” and “Grade 3 or more” TRAEs between the immunotherapy plus chemotherapy arm and the chemotherapy alone arm.
Then, figures display differences in several sub groups.
R3. Based on Figure 4, the text says “a statistically significant difference in terms of grade 3 or more hypertension was observed.” In recent years, data from meta-analyses have been increasingly emphasized. However, reliance solely on the statistical significance of p-values in interpreting such data has been a growing concern. Ideally, both p-values and minimal clinically important difference (MCID) should be considered. Unfortunately, MCID for effect sizes other than mean differences has not been well-established.
Recently, MCID applicable to effect sizes other than mean differences has been proposed as "Horita's MCID." Citing the following reference and incorporating a discussion on this topic, i.e. how you interpret OR=4.80, could further enhance the quality of this manuscript:
Horita N et al. Minimal clinically important difference (MCID) of effect sizes other than mean difference. J Clin Med 2024. DOI 10.69854/jcq.2024.0016.
https://jclinque.com/article/minimal-clinically-important-difference-mcid-of-effect-sizes-other-than-mean-difference
Thank you for the opportunity to review this excellent manuscript.
A3. In our analyses, we did not obtain statistically significant results either by considering the overall analyses or in the analyses stratified by specific irAE, either Any Grade or Grade >3, however, since some recent studies suggest that the minimal clinically important difference (MCID) should also be considered [REF: Horita et al. (2024)], this will be explored in the future.
We hope the revised paper will better suit the journal.
The Authors
